# HyperBatch: Scaling Contrastive Learning Batch Sizes by Two Orders of Magnitude

## Abstract

Contrastive learning has emerged as a powerful method for learning unsupervised representations of data that maximize similarity between "related" pairs of data and minimize similarity between unrelated pairs. Many contrastive losses depend heavily on the batch size, as larger batch sizes significantly improve model intelligence. However, modern backbones are memory intensive and limit the practical batch size one can train with. To alleviate this issue, we introduce a new framework to scale contrastive batch sizes by two orders of magnitude. This allows us to improve the performance of any contrastive learner. Our training framework consists of three phases—Pretrain, Adapt, and Fuse. In the Pretrain phase, we train a standard contrastive learner with conventional batch sizes. In the Adapt phase, we freeze the backbone and train a small number of later layers with very large batches, exposing these late-stage parameters to significantly larger batches and accelerated training. Finally, in the Fuse phase, we transfer large-batch adapter gradients back into the backbone with a modified version of backpropagation. We evaluate methods with audio-video contrastive learning on the Audioset dataset. We show that our multi-phase training pipeline significantly improves retrieval performance and outperforms baseline approaches in both speed and accuracy. By exposing the model to substantially more negatives we make each contrastive judgment orders of magnitude more challenging, encouraging models to develop more sophisticated and intelligent representations.

## 1 Introduction

Contrastive learning is one of the most effective methods to learn unsupervised representations of data across a wide variety of modalities. In contrastive learning, models learn to pull together similar, positive examples while pushing apart dissimilar, negative ones. Common contrastive loss functions, such as the InfoNCE (van den Oord et al., 2019), phrase this problem as a multi-way classification problem, where a model aims to distinguish positive pairs of data from negative pairs formed by combining random pairs of data in a batch. As one increases the batch size, one effectively increases the number of choices in this multi-class classification problem, boosting the probability of sampling hard negatives and yielding a much more challenging task. These especially challenging examples lead a model to develop more intelligent representations and help methods achieve state-of-the-art performance (Chen et al., 2020).

However, naively scaling the batch size of models comes with significant overhead, both in time and memory usage. Often, model creators are left with a difficult decision between scaling the model size, the batch size, or their number of GPUs (and hence the bill). Memory-intensive models that use contrastive methods face a key challenge: limited batch sizes directly restrict contrastive learning performance.

To address this limitation, we propose a method to scale contrastive batch sizes by over two orders of magnitude without requiring additional hardware. We introduce a novel multi-phase training framework – Pretrain, Adapt, and Fuse – that jointly trains a large backbone model and a lightweight contrastive head (Figure 1). The backbone and contrastive head are first pretrained by co-training on small batches to establish a strong joint initialization. Then, the contrastive head alone is trained with very large batches, up to two orders of magnitude larger than the initial batch size. We then calculate large-batch gradients from the contrastive head and propagate them in micro batches through the

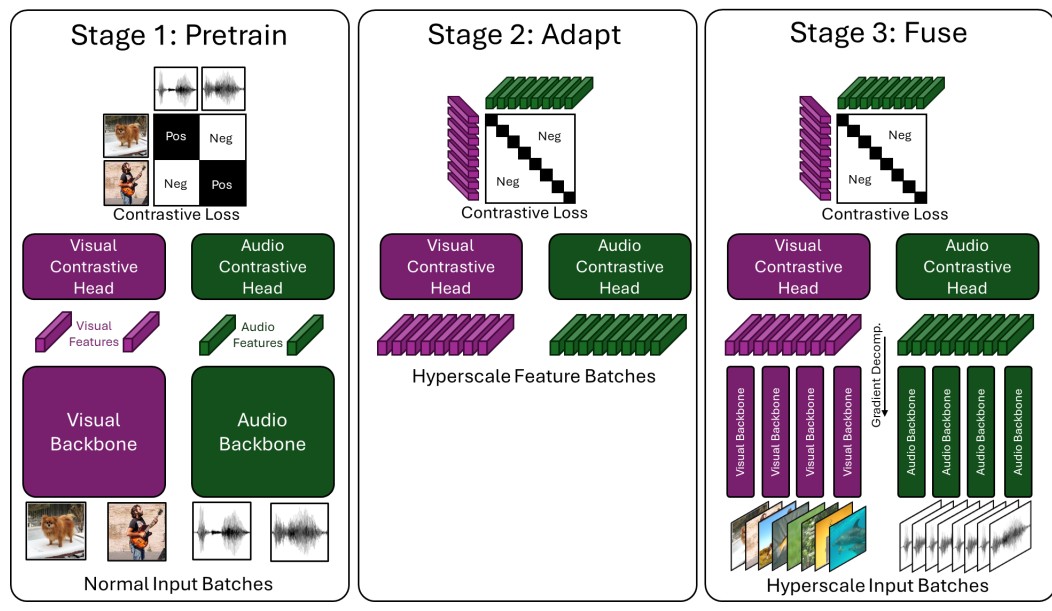

Figure 1: Phase-by-phase overview of the HyperBatch training framework.

backbone. This effectively fuses rich contrastive information into the backbone without exceeding memory limits. With this training framework, we multiply batch size by a scale of hundreds to thousands. This technique significantly expands the pool of negative pairs that the backbone learns from. We also leverage the lightweight nature of the contrastive head to reduce the speed of training while flooding training with "hard" negatives that sharpen the model. Our main contributions are as follows:

- We achieve large-batch benefits under the original memory budget by exposing the head to a vast negatives pool and distilling that signal back through the backbone.

- Our training framework accelerates contrastive learning by cheaply scoring massive batches, which translate to rapid accuracy gains.

- Our three phase design drops into any contrastive framework and backbone, requiring only a lightweight contrastive head.

## 2 RELATED WORK

**Contrastive Learning.** Contrastive learning has expanded significantly from vision-centric tasks to multimodal audio-visual learning. Early works exploited audio-visual synchrony without explicit supervision (Arandjelovic & Zisserman, 2017), inspiring cross-modal methods like XDC (Alwassel et al., 2020), which utilized pseudo-labeling from modality clustering. Modern methods (Morgado et al., 2021; Akbari et al., 2021; Nagrani et al., 2022; Guzhov et al., 2021; Girdhar et al., 2023) further advanced cross-modal retrieval and video understanding. Recently, DenseAV (Hamilton et al., 2024) introduced fine-grained spatial correspondences, achieving strong retrieval and segmentation performance by aligning dense audio-visual representations.

**Batch Size and Memory Efficiency.** Large batch sizes significantly enhance contrastive learning, as initially demonstrated by SimCLR (Chen et al., 2020). However, hardware constraints necessitate memory-efficient strategies. MoCo (He et al., 2020) introduced momentum encoders and memory queues, enabling large effective negative pools with small batch sizes. Other methods (Zbontar et al., 2021; Bardes et al., 2022; Caron et al., 2020) used novel loss objectives to prevent collapse without large batches or negative samples. Negative-free methods like BYOL and SimSiam (Grill et al., 2020; Chen & He, 2021) further reduced computational overhead. Additional memory op-

timization strategies include gradient accumulation, activation checkpointing (Chen et al., 2016a), and reversible networks (Chen et al., 2017).

**Adapters for Parameter Efficiency.** Adapters have emerged as a lightweight method for efficient training. Early adapter modules (Sylvestre-Alvise Rebuffi, 2017) integrated small, task-specific residual layers within frozen models. Subsequent methods, such as LoRA (Hu et al., 2022) and IA³ (Liu et al., 2022), exploited low-rank decompositions and scalar parameterizations, significantly reducing memory and computational costs. Recent variants have effectively adapted visual and audio-visual transformers (Chen et al., 2022; Gao et al., 2025; Jia et al., 2022; Lin et al., 2023; Abduljalil Radman, 2023; Wang et al., 2024), facilitating efficient multimodal contrastive training. Generally, adapter-like modules can add only a few trainable objectives (Mahmud et al., 2024; Duan et al., 2023; Cheng et al., 2024) to inject cross-modal interactions into contrastive learning frameworks.

**Teacher-Student Frameworks.** Teacher-student architectures provide stable training signals by leveraging slowly updated teacher networks, minimizing reliance on negatives. Methods such as MoCo and BYOL (He et al., 2020; Grill et al., 2020) demonstrated robust embeddings with minimal negative sampling. Semi-supervised variants like Noisy Student (Xie et al., 2020) utilized iterative pseudo-labeling processes. Subsequent extensions using vision transformers (Caron et al., 2021; Zhou et al., 2022; Chen et al., 2021a; Oquab et al., 2024) and multimodal tasks (Radford et al., 2021; Li et al., 2021; Chen et al., 2021b) illustrate broad applicability and effectiveness across diverse computational setups.

## 3 METHODS

We propose a three-phase training framework–Pretrain, Adapt, and Fuse–which progressively optimizes the backbone and contrastive head with each phase. Our method fine-tunes later layers on huge batches and then tranfers that signal into the backbone. This enables the backbone to reach large-batch performance without directly training on larger batches.

### 3.1 MODEL ARCHITECTURE

The model architecture consists of two primary components:

1. Memory-Intensive Backbone: A model that extracts representations from audio and visual inputs.

2. Contrastive Head: A lightweight module consisting of multi-layer perceptron (MLP) to improve on the backbone's audiovisual representations.

The backbone uses parallel image and audio featurizers to generate class tokens. HuBERT (Hsu et al., 2021) is used as the audio backbone, and DINO (Caron et al., 2021) is used as the image backbone. In both the audio and visual branches, we then channel-normalize the class token and apply a linear layer to bring the class tokens into a joint embedding space and reduce training instability.

The contrastive head applies parallel MLPs to the features generated by the backbone. These residual modifications are added to the original features from the backbone. In this way, these later layers learn corrective adjustments that optimize for the contrastive learning objective, refining the backbone's representations.

The full model output can be represented as

$$\mathbf{z} = y + g_\theta(y), \qquad y = f_\phi(\mathbf{x})$$

where $\mathbf{x}$ represents the input data, $f_\phi$ denotes the backbone with parameters $\phi$, and $g_\theta$ represents the contrastive head with parameters $\theta$. The final output $\mathbf{z}$ is used to compute a contrastive loss $\mathcal{L}$. The total loss aggregates the InfoNCE loss (van den Oord et al., 2019) of retrieving audio from image with the loss from retrieving image from audio.

## 3.2 FRAMEWORK

We outline the forward and backward propagation procedures for each of the phases below.

### 3.2.1 PRETRAIN

In the first phase, we jointly warm-start the backbone $f_\phi$ and contrastive head $g_\theta$ under the small-batch memory budget, establishing initial representations and a stable training configuration. Each Pretrain step uses standard backpropagation techniques on a single small batch.

**Forward pass.** For a small batch $x$,

$$y = f_\phi(x), \qquad z = y + g_\theta(y), \qquad L = \mathcal{L}(z),$$

where $\mathcal{L}$ is the weighted sum of the audio-to-image and image-to-audio InfoNCE losses.

**Backward pass.** The MLP parameters $\theta$ and the backbone parameters $\phi$ are updated as

$$\nabla_\theta L = \frac{\partial L}{\partial z}\frac{\partial z}{\partial \theta},$$

$$\nabla_\phi L = \frac{\partial L}{\partial z}\frac{\partial z}{\partial y}\frac{\partial y}{\partial \phi}.$$

### 3.2.2 ADAPT

This phase exploits very large batch sizes to learn strong contrastive structure in the head while keeping memory bounded by freezing the backbone and training only $g_\theta$ on backbone-featurized data.

We construct a large effective batch $X = \{x_k\}_{k=1}^n$ and featurize with the Pretrain-trained backbone to obtain

$$Y = \mathrm{concat}\big(f_\phi(x_1), \ldots, f_\phi(x_n)\big).$$

To further increase speedup during this phase, $Y$ is precomputed and cached.

**Forward pass.** The MLPs produce

$$Z = Y + g_\theta(Y), \qquad L = \mathcal{L}(Z).$$

**Backward pass.** With $\phi$ frozen and $Y$ treated as constant, $\theta$ can be updated as

$$\nabla_\theta L = \frac{\partial L}{\partial Z}\frac{\partial Z}{\partial \theta} \quad , \text{ with } f_\phi(X) \text{ fixed and } \phi \text{ remaining frozen.}$$

### 3.2.3 FUSE

Fuse transfers large-batch contrastive information gathered by the contrastive head back into the memory-limited backbone. We do this by computing the loss and gradients on a large effective batch, then propagating the resulting intermediate gradients through the backbone in small microbatches. This mechanism transforms the conventional training pipeline into a hybrid model, transferring the head's large-batch learning to the backbone in a memory-efficient manner. We pick up the state of the backbone from the Pretrain phase and the state of the MLPs from the Adapt phase.

**Forward pass.** Let $X = \{x_k\}_{k=1}^n$ be a large batch partitioned into $n$ microbatches. We sequentially featurize $x_k$ through $f_\phi$ and concatenate to form $Y$:

$$Y = \mathrm{concat}(f_\phi(x_1), ..., f_\phi(x_k)).$$

We then compute $Z$ and $L$:

$$Z = Y + g_\theta(Y), \qquad L = \mathcal{L}(Z).$$

**Backward pass: contrastive head and intermediate gradient.** We first update the contrastive head on the large batch and compute the gradient w.r.t. the large-batch backbone output $Y$:

$$\nabla_\theta L \;=\; \frac{\partial L}{\partial Z}\frac{\partial Z}{\partial \theta}, \qquad \nabla_Y L \;=\; \left(\frac{\partial Z}{\partial Y}\right)^\top \nabla_Z L.$$

For the contrastive head $Z = Y + g_\theta(Y)$,

$$\frac{\partial Z}{\partial Y} \;=\; I + J_{g_\theta}(Y), \quad \Rightarrow \quad \nabla_Y L \;=\; \nabla_Z L \;+\; J_{g_\theta}(Y)^\top \nabla_Z L,$$

where $J_{g_\theta}(Y)$ is the Jacobian of $g_\theta$ at $Y$.

**Backward pass: backbone in microbatches.** We distribute $\nabla_Y L$ across the concatenated slices to obtain per-microbatch signals $\{\nabla_{y_k} L\}_{k=1}^n$ aligned with $y_k = f_\phi(x_k)$. For each microbatch $k = 1, \ldots, n$ we run a backward pass through $f_\phi$:

$$\nabla_\phi L^{(k)} \;=\; \left(\nabla_{y_k} L\right)^\top \frac{\partial y_k}{\partial \phi}.$$

At a high level, within each Fuse iteration over $X$ we perform:

1. Large-batch forward: build $Y$, compute $Z$, $L$.
2. Contrastive head step: backpropagate $L$ to update $\theta$ and obtain $\nabla_Y L$.
3. Backbone micro-steps: for $k = 1 \ldots n$, backpropagate $\nabla_{y_k} L$ through $f_\phi$ to update $\phi$; accumulate gradients to an effective batch of $|X|$.

We avoid reusing stale $\nabla_Y L$ across multiple backbone updates: the MLPs are updated once per large batch (Step 2), after which the backbone is updated exactly once via the $n$ microbackward passes (Step 3). This minimizes staleness while respecting memory limits.

**Gradient Checkpointing** We utilize gradient checkpointing (Chen et al., 2016b) to streamline the microbatch forward and backward passes. Microbatch forwards through $f_\phi$ are wrapped in activation checkpoints; during Step 3, each $\nabla_{y_k} L$ triggers a recomputation of the $k$-th microbatch activations for $f_\phi$ only, keeping memory bounded while preserving exact gradients.

### 3.3 TRAINING

We train and experiment with our three-phase approach on the AudioSet Dataset (Gemmeke et al., 2017). Our image processing pipeline is as follows: We randomly sample a single video frame and apply random resizing, color jittering, random grayscaling, and random flipping as augmentations on the training dataset. All audio is first resampled to 16KHhZ to be compatible with HuBERT. We then apply a series of augmentations: additive noise, gain perturbation, pitch shift, and reverberation. We train on 4 A100 GPUs. We use a micro batch size of 40 for the Pretrain phase, and we experiment with batch sizes ranging from 1000 to 50,000 for the Adapt phase. For the Fuse phase, we use a large batch size of 1000, and a micro batch size of 40. A comparison of different batch sizes used in the Adapt phase are in Table 2.

## 4 RESULTS

We evaluate on cross-modal retrieval using 1,000 clips derived from AudioSet. Given each sample in one modality, we rank all 1,000 candidates from the opposite modality. We report Recall@k for $k = 1, 5, 10$, which is the percentage of queries whose ground-truth pair ranked in the top-k. Image-to-audio recall ranks audio clips given a fixed image, and audio-to-image recall ranks images given a fixed audio clip.

Figure 2 illustrates the R@1 trajectories across the three training phases. Our approach yields a clear boost in performance at each new training phase. As a training phase converges, the subsequent phase drives the model to a new spike in performance. Table 1 reports the best R@k achieved in each phase under a fixed total of 130,000 steps. The total step count breaks down to 95k steps for Pretrain, 25k steps for Adapt, and 10k steps for Fuse. Improvements in parentheses are relative to Pretrain. We also report recall against a baseline to highlight gains over gradient accumulation.

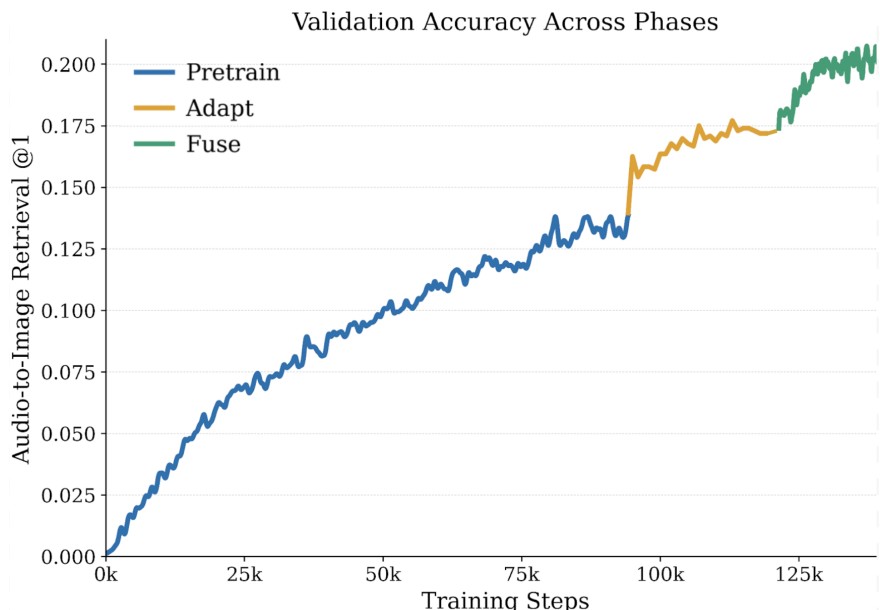

Figure 2: Top-1 audio-to-image validation recall across phases.

Table 1: Image-to-audio and audio-to-image retrieval accuracies on the AudioSet dataset.

| $I \rightarrow A$ | Pretrain | Adapt | Fuse | Baseline |
|---|---|---|---|---|
| R@1 | 16.67 | 18.93 (**+2.26**) | 24.58 (**+7.91**) | 22.52 |
| R@5 | 38.85 | 41.04 (**+2.19**) | 46.25 (**+7.40**) | 44.60 |
| R@10 | 49.06 | 51.56 (**+2.50**) | 55.42 (**+6.36**) | 54.13 |

| $A \rightarrow I$ | Pretrain | Adapt | Fuse | Baseline |
|---|---|---|---|---|
| R@1 | 13.96 | 17.19 (**+3.23**) | 22.60 (**+8.64**) | 20.02 |
| R@5 | 35.73 | 39.38 (**+3.65**) | 44.27 (**+8.54**) | 42.52 |
| R@10 | 47.19 | 51.46 (**+4.27**) | 53.54 (**+6.35**) | 52.35 |

## 4.1 GRADIENT ACCUMULATION COMPARISON

As a strong baseline, we implement gradient accumulation across microbatches. The InfoNCE loss is computed separately on each microbatch, and the optimizer is stepped once every $k$ microbatches such that the effective batch size matches the huge batch size used in the Adapt and Fuse phases. With microbatch size $b = 40$ under backbone memory constraints and target effective batch $B = 1000$, we set $k = 25$ accumulation steps, so the optimizer updates once per $k$ microbatches. Training for the baseline resumes from the same Pretrain checkpoint as Adapt picks up from.

In this setting, each microbatch forms a $b \times b$ similarity matrix and contributes $b^2 - b$ negative pairs; across the $k$ microbatches this totals to $k(b^2 - b)$ negatives per step. In the MLP updates of the Adapt and Fuse phases, we assemble a single $B \times B$ similarity matrix over the full effective batch, yielding $B^2 - B$ negatives per step. The exact ratio of negative pairs per optimizer step is

$$\frac{B^2 - B}{k(b^2 - b)} = \frac{kb - 1}{b - 1} = k + \frac{k - 1}{b - 1}.$$

This ratio that our negative pool grows by is strictly greater than $k$, and even larger when $k > b$. Geometrically, the baseline covers only the $k$ diagonal $b \times b$ blocks of the full $B \times B$ matrix, whereas our method uses the entire matrix including off-diagonal interactions (Figure 3). This loss-time exposure to off-diagonal interactions, not achievable by accumulation, is the key driver of the consistent gains in Table 1.

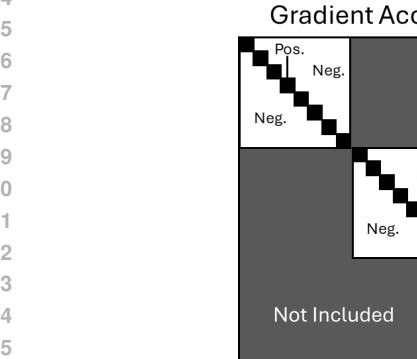 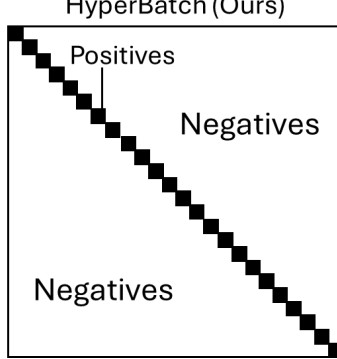

Figure 3: Visual comparison of negative pool between Hyperbatch and gradient accumulation.

Table 2: Comparison of batch sizes used in the Adapt phase.

| Batch Size | R@1 | R@5 | R@10 | Wall Time |
|---|---|---|---|---|
| 1000 | 18.93 | 41.04 | 51.56 | 1h 16m |
| 10,000 | 19.47 | 42.16 | 53.01 | 4h 31m |
| 50,000 | 19.83 | 42.37 | 53.50 | 1d 9h 22m |

## 4.2 ADAPT PHASE BATCH SIZE

We experiment with Adapt-phase batch sizes from 1000 to 50,000 (i.e., 25 to 1250 times larger than the Pretrain phase micro batch size $b = 40$). Table 2 reports image to audio recall accuracy and wall time. All runs resume from the same Pretrain checkpoint at 95k steps.

Increasing the Adapt batch size $B$ yielded consistent, though sub-linear, accuracy gains. This aligns with the saturation of the InfoNCE partition function as negatives become plentiful. In practice, $B = 10k$ offers a favorable accuracy to time compromise, while $B = 50k$ maximizes accuracy under a longer wall time.

The accuracy gains from increasing batch size capture how our Pretrain-Adapt-Fuse design allows us to materialize very large-batch contrastive losses. The Adapt phase circumvents memory constraints by computing the InfoNCE objective over a true large batch at the level of the contrastive head, where the gradients are memory-cheap. Unlike memory-bank or queue approaches, Adapt uses the contrastive head to score a single-step, in-batch objective over the entire large batch and avoids staleness from long-lived keys. Unlike pure gradient accumulation, Adapt's negative set is the full batch at loss time, exposing the head to a much dense hard-negative distribution.

## 5 CONCLUSION

We introduce **Pretrain–Adapt–Fuse**, a three-phase training framework that makes abundant hard-negative sampling practical for contrastive learning under strict memory budgets. Our framework leverages the idea of decoupling where large batches are scored from where gradients are stored. In **Pretrain**, we co-train a memory-intensive backbone with a lightweight head on small batches to establish stable features and a strong initialization. In **Adapt**, we freeze the backbone and train only the head on true large batches—scaling batch size by two orders of magnitude to expose the model to a much dense distribution of hard negatives. Finally, in **Fuse**, we push the large-batch signal back into the backbone through modified backpropagation. In doing so, we transfer large-batch structure into a memory-constrained backbone.

Empirically, this framework boosts retrieval accuracy at each phase on AudioSet. Relative to Pretrain, Adapt and Fuse deliver consistent gains from where the Pretrain phase converges, with R@1 improving 16.67 to 24.58 for image to audio retrieval and 13.96 to 22.60 for audio to image retrieval

under a fixed total step budget. We also find that increasing Adapt batch size monotonically improves accuracy, matching the intuition that more negatives increase the chance of sampling harder ones. Our method outperforms a strong gradient-accumulation baseline matched for effective batch size by capturing off-diagonal negative pairs that accumulation never sees within the same step.

Practically, our three phase training framework is simple to integrate and broadly applicable: it treats the classification head as a low-memory "contrastive amplifier" for large-batch learning, then fuses that signal back to the backbone. By scaling batch size by hundreds to thousands without enlarging the backbone's activation footprint, we enable richer, harder negative sampling and faster accuracy gains per optimizer step. The result is a drop-in training scheme that preserves the benefits of massive contrastive batches while respecting memory constraints, that can be used across contrastive frameworks and modalities.

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
