# OpenReview forum: "HyperBatch: Scaling Contrastive Learning Batch Sizes by Two Orders of Magnitude"
_ICLR.cc/2026/Conference — ICLR 2026 Conference Withdrawn Submission_

### Official Review · Reviewer_CMkJ · 2025-10-29

**Soundness:** 1
**Presentation:** 2
**Contribution:** 1
**Rating:** 2
**Confidence:** 5

**Summary:**

The authors propose HyperBatch, a three-phase training framework for scaling contrastive learning batch sizes by two orders of magnitude without additional memory requirements. Two key components are introduced: a memory-intensive backbone for feature extraction and a lightweight contrastive head for representation refinement. The framework consists of three phases: Pretrain for joint initialization with small batches, Adapt for training the head alone on large batches (1,000-50,000 samples) using cached backbone features, and Fuse for transferring large-batch gradients back to the backbone through microbatch-wise backpropagation. For gradient computation, gradient checkpointing is utilized for memory efficiency, and the InfoNCE loss is computed over the full concatenated batch rather than per-microbatch. The experimental results on AudioSet show improvements in audio-visual retrieval metrics compared to gradient accumulation baselines.

**Strengths:**

1. The paper correctly identifies that large batch sizes are crucial for contrastive learning performance, as they provide more negative samples and harder negatives that lead to better representations. Memory constraints that limit batch sizes are indeed a significant practical bottleneck in training large-scale contrastive models, and attempting to address this limitation is a valuable research direction.

2. While methods like MoCo, BYOL, and SimSiam have primarily focused on single-modality (vision) tasks, this work attempts to tackle the more challenging multi-modal setting with audio-visual contrastive learning. Multi-modal representation learning is increasingly important for real-world applications, and exploring memory-efficient training methods in this context is commendable.

**Weaknesses:**

1. Methodological inconsistency: The core claim of achieving large-batch training is questionable. During the Adapt phase, the backbone remains frozen and only the lightweight head is trained on cached, static features, which is fundamentally different from true large-batch contrastive learning where the backbone parameters are updated with gradients computed from large batches, such as MoCo. The approach more closely resembles post-hoc feature refinement rather than large-batch training.

2. Unclear gradient transfer mechanism: The Fuse phase attempts to backpropagate gradients from a head trained on frozen features (from Adapt phase) to update the current backbone. This creates a mismatch between the feature distribution the head was optimized for and the current backbone's output distribution. The theoretical justification for why gradients computed on outdated feature distributions should provide meaningful updates to the current backbone is absent.

3. Insufficient experimental validation: The evaluation is limited to a single dataset (AudioSet) and a single task (cross-modal retrieval). The paper lacks evaluation on well-established benchmarks such as ImageNet for visual representation learning, which would allow direct comparison with existing large-batch contrastive learning methods. The paper also lacks comparison with established memory-efficient contrastive learning methods such as MoCo, BYOL, or SwAV. Additionally, there are no downstream task evaluations to demonstrate whether the improvements in retrieval metrics translate to better representations for practical applications.

4. Unfair baseline comparison: The gradient accumulation baseline computes InfoNCE loss per microbatch, while the proposed method computes it over the full concatenated batch. This difference in loss computation, rather than the three-phase framework itself, could account for the observed improvements. A fair comparison would require both methods to use identical loss computation strategies.

5. Missing technical details and ablation studies: Critical implementation details are absent, including learning rate schedules for different phases, optimization hyperparameters, and the specific architecture of the contrastive head. The paper lacks ablation studies to identify which components contribute to performance gains. The asymmetric training steps (95k for Pretrain, 25k for Adapt, only 10k for Fuse) raise questions about convergence and optimal phase duration.

6. Limited theoretical analysis: The paper provides no formal analysis of convergence properties or theoretical guarantees that the proposed method approximates true large-batch training. The relationship between the solution obtained through this three-phase approach and that of standard large-batch training remains unclear.

**Questions:**

1. Clarification on the core training mechanism: Could you provide a mathematical derivation showing how gradients computed from a head trained on frozen features (Adapt phase) provide valid optimization directions for the current backbone (Fuse phase)? Specifically, during Adapt, the head learns to process features from a frozen backbone with a fixed distribution. However, during Fuse, this same head is applied to features from an updated backbone that has evolved through training. How would you address this distribution shift between training and application of the head? What is the theoretical or empirical justification that the head's learned transformations on static features remain beneficial when applied to the evolving feature distribution during Fuse?

2. Regarding the baseline comparison: The evaluation in Section 4.1 appears to conflate the training framework's contribution with the loss computation strategy's contribution. The paper states that the baseline computes the InfoNCE loss separately on each microbatch, forming $b \times b$ similarity matrices. In contrast, the proposed method assembles "a single $B \times B$ similarity matrix over the full effective batch". To accurately isolate the performance gains attributable to the ''Pretrain-Adapt-Fuse'' framework itself, distinct from the known benefits of a larger negative pool, could the authors provide results for a baseline that also computes the loss over the full $B \times B$ concatenated batch?

3. Ablation studies: What happens if you: (a) continue training the backbone jointly with the head in Adapt instead of freezing? (b) use different ratios of training steps across phases? These ablations would help identify which components are essential.

4. Comparison with established methods: How does HyperBatch compare with MoCo, BYOL, and SimSiam? Could you also provide results on standard benchmark datasets like ImageNet to enable direct comparison with existing large-batch contrastive methods?

5. Convergence analysis: Why does the Fuse phase only run for 10k steps compared to 95k for Pretrain? Could you provide training curves showing loss/accuracy throughout all three phases?

6. Downstream task evaluation: Do the improvements in retrieval metrics translate to better performance on downstream tasks (e.g., classification, detection)? This would validate whether the learned representations are generally better or just optimized for retrieval.

7. Technical specifications: Could you provide the exact architecture of the contrastive head, learning rates for each phase, and optimizer settings? This information is crucial for reproducibility.

---

### Official Review · Reviewer_ZkXL · 2025-10-31

**Soundness:** 2
**Presentation:** 1
**Contribution:** 2
**Rating:** 2
**Confidence:** 2

**Summary:**

This paper proposes HyperBatch, a framework to overcome memory limits when training large contrastive learning models. It uses a three-phase (Pretrain, Adapt, Fuse) approach where a lightweight head is first trained on massive batches, and then a novel backpropagation step fuses this large-batch gradient information back into the memory-constrained backbone, achieving the benefits of massive batches without the high memory cost.

**Strengths:**

* The paper addresses an extremely important and common practical bottleneck in contrastive learning: how to effectively scale batch sizes under strict memory budgets.
* The proposed three-phase (Pretrain-Adapt-Fuse) framework is novel, cleverly decoupling the large-batch loss computation (in a lightweight head) from the gradient updates of the memory-intensive backbone.

**Weaknesses:**

1. The framework's core is the Pretrain-Adapt-Fuse pipeline. However, the necessity of the separate Adapt phase is not fully ablated. The Fuse phase already trains the head on large batches. It is unclear if the Adapt phase is essential for stability or if a simpler Pretrain → Fuse pipeline would achieve comparable results. An ablation study removing the Adapt phase would clarify the framework's essential components.

2. The method is demonstrated on a single audio-visual retrieval task. The paper claims it is a "drop-in training scheme" applicable to any contrastive framework, but this broad claim is not substantiated. It would be significantly more convincing to see results on standard uni-modal benchmarks, such as SimCLR on ImageNet, to prove that the framework is truly general-purpose and not just tailored to the specific audio-visual architecture used.

**Questions:**

1. Why was gradient accumulation (GA) chosen as the primary baseline rather than momentum encoders (such as MoCo) which also solve the large batch problem? What are the advantages of HyperBatch compared to MoCo?

2. Is the Adapt phase necessary? What would happen if we skipped the Adapt phase and went directly from Pretrain to Fuse?

3. You claim this is an "off-the-shelf" solution, but experiments are limited to audio-visual tasks. Have you tested the method on standard uni-modal benchmarks such as SimCLR on ImageNet to demonstrate its generalizability?

---

### Official Review · Reviewer_ESoe · 2025-11-01

**Soundness:** 2
**Presentation:** 2
**Contribution:** 2
**Rating:** 4
**Confidence:** 3

**Summary:**

This paper proposes a three-phase contrastive learning framework — Pretrain, Adapt, and Fuse — to scale effective contrastive batch sizes by two orders of magnitude without additional hardware costs. The approach freezes the backbone after initial training and then trains a lightweight contrastive head on extremely large batches, later fusing gradients back into the backbone in a micro-batching manner. The authors claim improved performance and faster training on AudioSet by exposing the model to many more hard negatives than would otherwise fit in memory.

**Strengths:**

1. This paper tackles a practical and important limitation in contrastive learning—namely, the difficulty of scaling effective batch sizes due to memory constraints—and presents a simple, modular multi-phase training strategy to overcome it. The proposed Pretrain–Adapt–Fuse pipeline is clearly described, easy to integrate into existing contrastive frameworks, and does not require specialized hardware or distributed setup, making it accessible to practitioners.

2. The method takes advantage of freezing a backbone and training a lightweight contrastive head on very large batches, enabling exposure to many more hard negatives and improving representation quality. The authors motivate the approach well, provide intuitive reasoning about its benefits compared to naïve gradient accumulation, and demonstrate performance improvements on AudioSet retrieval tasks. The writing is clear and the implementation concepts are straightforward, which increases the practical value of the paper.

**Weaknesses:**

1. Novelty. While useful in practice, the contribution is conceptually incremental—it largely combines established training tricks such as backbone freezing, micro-batch gradient accumulation, and staged optimization, rather than introducing fundamentally new contrastive learning theory or algorithms.

2. Empirical evidence. The experimental validation is limited in scope, relying primarily on AudioSet without evaluation on standard large-scale vision benchmarks (e.g., ImageNet, CLIP-style settings) or across diverse architectures, raising questions about generality and scalability. Important baselines such as memory-bank approaches (e.g., MoCo queues) and distributed large-batch training are absent, making it difficult to isolate the benefit of the proposed method relative to modern contrastive systems.

3. Ablation analysis. The paper does not analyze potential degradation from freezing early layers, or efficiency trade-offs versus simply training longer or adding compute. The argument that off-diagonal negatives drive improvement is plausible but not rigorously demonstrated, giving the work a somewhat heuristic and engineering-driven character rather than a theoretically grounded advancement.

**Questions:**

Please see weakness for questions.

---

### Official Review · Reviewer_g5wG · 2025-11-01

**Soundness:** 2
**Presentation:** 2
**Contribution:** 2
**Rating:** 2
**Confidence:** 3

**Summary:**

This paper introduces HYPERBATCH, a method for scaling contrastive learning to batch sizes two orders of magnitude larger than conventional approaches. While large batches are known to benefit contrastive objectives, modern backbone models make such scaling impractical due to memory constraints.
The authors propose a three-step training procedure in which only selected parts of the model are trained on large batches, with gradients transferred through a modified backpropagation mechanism.
Experimental results suggest that HYPERBATCH achieves faster convergence and higher accuracy than baseline methods.

**Strengths:**

- The paper presents an interesting and practical idea addressing an important scalability limitation in contrastive learning.
- It is well-written and easy to follow, with clear exposition of the method.
- Figure 1 summarizes the proposed approach and helps with conceptual understanding.

**Weaknesses:**

- The paper does not discuss limitations of the approach, such as potential instability or applicability constraints.
- There is no released code or detailed training configuration, which limits reproducibility.
- The absence of an ablation study makes it difficult to assess which components of the proposed method drive the reported gains.

**Questions:**

- Why is the proposed approach evaluated only in the context of contrastive learning? The method appears potentially applicable to other paradigms involving large batch training.
- Why are experiments limited to audio–video pairs? Could the authors extend or at least discuss applications to other modalities (e.g., image–text or cross-language)?

---

### Note · Authors · 2025-12-03

**Comment:**

We sincerely thank the reviewers for their feedback and suggestions, which we will use to improve this work in future iterations.

**Withdrawal Confirmation:**

I have read and agree with the venue's withdrawal policy on behalf of myself and my co-authors.